# Prevalence and risk factors of uncorrected refractive error among an elderly Chinese population in urban China: a cross-sectional study

Hehua Ye,[1] Yiyong Qian,[2] Qi Zhang,[1] Xiaohong Liu,[3] Xuan Cai,[3] Wenjing Yu,[1] Xiang Li,[1] Peiquan Zhao[1]

HY and YQ contributed equally.

[1]Department of Ophthalmology, Xinhua Hospital, Shanghai Jiao Tong University, Shanghai, China
[2]Department of Ophthalmology, Shanghai Tenth People's Hospital, Tongji University School of Medicine, Shanghai, China
[3]Shanghai Jiao Tong University Affiliated Sixth People's Hospital, Department of Ophthalmology, Shanghai, China

**Correspondence to**
Dr Peiquan Zhao;
zhaopeiquan@126.com

## ABSTRACT

**Objectives** To investigate the prevalence and risk factors of uncorrected refractive error (URE) in an elderly urban Chinese population in China.

**Design** A population-based cross-sectional study.

**Methods** The study was conducted using a cluster random sample of residents aged 50 years or older living in the Jiangning Road subdistrict, Shanghai, China. All participants underwent a standardised interview and eye examinations, including presenting visual acuity (PVA) and best-corrected visual acuity (BCVA) between November 2012 and February 2013. URE was defined as an improvement of two lines or more in the BCVA compared with the PVA in the better eye of <20/40.

**Results** A total of 1999 subjects (an 82.5% response rate) completed both the questionnaire and ophthalmic examination. The prevalence of URE was 20.1% (95% CI 18.0% to 22.2%) in the study sample. After age standardisation, the prevalence of URE in Chinese people aged 50 years or older was 18.7% (95% CI 17.0% to 20.4%). Under multiple logistic regression analysis, older age (per 1-year increase, OR 1.04, 95% CI 1.03 to 1.05) and a lower level of education (OR 1.34, 95% CI 1.07 to 1.69) were significantly related to URE. A history of ocular diseases (OR 0.71, 95% CI 0.55 to 0.92) was a protective factor for URE.

**Conclusions** URE is highly prevalent among the elderly urban Chinese population, which should raise awareness of the URE burden in China to meet the Vision 2020 goal to eliminate preventable blindness.

### Strengths and limitations of this study

► A high response rate in a large population-based sample.
► Standardised protocols based on the typical definition of uncorrected refractive error (URE).
► Uncorrected near vision, that is, presbyopia, has not been evaluated in this population.
► The underlying reasons of URE highly prevalent among elderly urban Chinese remains unknown.

from vision impairment due to URE, especially older persons. Improvement in the vision-dependent quality of life of older persons has been demonstrated when URE is corrected.[4 5]

A wide variation in the prevalence of URE worldwide has been reported.[6] In East Asian countries, the prevalence of URE is potentially higher due to a higher prevalence of refractive errors.[7] Mainland China comprises one-fifth of the world's population with 78 million people aged 60 years and above, and a substantial increase in the number of older persons is expected in the next few decades.[8] Despite the potential magnitude of this problem, there have been few population-based studies on URE in older persons in mainland China.[9 10]

The purpose of the present study was to describe the prevalence and risk factors of URE in an elderly population in Shanghai, which is the largest city by population in China. The findings of this study may be helpful in determining the strategy to meet the Vision 2020 goal to eliminate preventable blindness.

## INTRODUCTION

Uncorrected refractive error (URE) is the most common cause of vision impairment and the secondary cause of blindness worldwide.[1] It has been estimated that URE accounts for 153 million individuals of visual impairment globally, and the WHO identified URE as one of the priorities for the programme of Vision 2020.[1]

URE is associated with limitations in vision-related tasks and decreased quality of life.[2 3] Despite the relatively easy intervention for refractive error, many people still suffer

## METHODS
### Study population

The Jiangning Eye Study, a population-based cross-sectional study of urban Chinese elders aged 50 years and older living in the Jiangning Road, Jing'an District, Shanghai, was conducted

to assess the prevalence and risk factors of ocular diseases. The study design and details of population sampling have been described elsewhere.[11] [12] The study followed the guidelines in the Declaration of Helsinki. Informed written consent was obtained from each participant.

## Study procedures

An interviewer-administered questionnaire was conducted to gather information about each participant's demographics, lifestyle (eg, cigarette smoking and alcohol consumption), socioeconomic status factors (eg, marital status, income level and final education level), medical history and history of ocular diseases. The ocular examination was conducted according to a standardised protocol included the presenting visual acuity (PVA) and best-corrected visual acuity (BCVA), autorefraction and subjective refraction, non-contact tonometry, slit-lamp biomicroscopy and indirect ophthalmoscopy. Distance visual acuity was assessed using Early Treatment Diabetic Retinopathy Study charts and was recorded separately for each eye. All participants were asked to bring their spectacles before performing ocular examinations. The PVA was measured with the subject's spectacles. If the participants did not wear spectacles or did not bring their spectacles, the PVA was measured without spectacles. If the PVA was <20/20, the BCVA was assessed with subjective refraction.

## Definitions

URE was defined as an improvement of two lines (0.2 logMAR) or more in the BCVA compared with the PVA in the better eye of <20/40.[13] [14] Refractive status was expressed using the spherical equivalent (SE; sphere +1/2 cylinder) calculated from the BCVA. An SE between −1.0 and +1.0 D was defined as emmetropia, an SE <−1.0 D as myopia and an SE >+1.0 as hyperopia.

## Statistical analysis

The overall prevalence (%) of URE was calculated. The age-standardised prevalence was calculated using direct standardisation of the study samples to the 2010 Chinese population census.[8] The correlation between the refractive status of the right and left eyes was calculated using the Spearman correlation coefficient. A multiple logistic regression analysis was assessed with URE as the dependent variable. The relevant predictors were used as the covariates. Statistical analysis was performed with the Statistical Package for Social Science (SPSS V.15.0) software. A P value less than 0.05 was considered statistically significant.

## RESULTS

### Participants and descriptive data

Of 2478 eligible participants identified for the Jiangning Eye Study, 2044 (82.5% response rate) underwent ocular examinations in a temporary clinic. Data from 1999 subjects who completed both the questionnaire and ophthalmic examination were included and analysed in the present study.

The mean age (±SD) was 64.7 (±9.9) years, and 56.2% were women. The age distribution of the population was 50–59, 757 (37.9%); 60–69, 672 (33.6%); 70–79, 352 (17.6%); and 80 years or older, 218 (10.9%). The high correlation was found between the right and left eyes of refractive status (r=0.83; P<0.001). Of the participants, 30.6% were myopic and 39.6% were hyperopic in the right eye.

## The prevalence of URE

The prevalence rate of URE in the Jiangning Eye Study is summarised in table 1. The crude prevalence rate of URE in the entire study sample was 20.1% (95% CI 18.0% to 22.2%). After age standardisation to the 2010 Chinese population census, the prevalence of URE in Chinese people aged 50 years or older was estimated to be 18.7% (95% CI 17.0% to 20.4%). There was a significant age-related trend in the prevalence of URE in the entire study sample (P value for the trend was <0.001). No significant difference was found between men (19.3%) and women (20.7%) in the prevalence rate of URE ($X^2$=0.613, P=0.434). Among those who wore spectacles or contact lenses (only two participants wore contact lenses), 16.5% were still uncorrected (ie, a gain of two or more lines). Among the 101 participants with prior cataract surgery (at least one eye), 17.8% were uncorrected.

## Analysis of associated factors

Table 2 summarises the age-adjusted and multivariate-adjusted logistic regression model of the predictors for URE. URE was significantly associated with older age (per 1-year increase, OR 1.05, 95% CI 1.04 to 1.06). After adjusting for age, a lower level of education (OR 1.36, 95% CI 1.09 to 1.70) was a significant risk factor for URE. On the other hand, a history of ocular disease (OR 0.71, 95% CI 0.55 to 0.91) was a significant protective factor for URE. In the final multiple logistic regression analysis, older age (per 1-year increase, OR 1.04, 95% CI 1.03 to 1.05) and a lower level of education (OR 1.34, 95% CI 1.07 to 1.69) were still significantly associated with URE, whereas a history of ocular disease (OR 0.71, 95% CI 0.55 to 0.92) was negatively associated with URE.

## DISCUSSION

### Key results

This population-based study provides novel data on the prevalence of URE in an elderly urban Chinese population in China. URE was present in 20.1% of the study sample, defining URE as an improvement of two lines (0.2 logMAR) or more in the BCVA compared with the PVA in the better eye of <20/40. Older age and a lower level of education were significantly related to URE, whereas a history of ocular diseases was a protective factor.

**Table 1** Prevalence rates of uncorrected refractive error by gender and age in the Jiangning Eye Study

| Age group (years) | All | | | Spectacle wearers | | | Non-spectacle wearers | | |
|---|---|---|---|---|---|---|---|---|---|
| | N | n | Prevalence rate (%) | N | n | Prevalence rate (%) | N | n | Prevalence rate (%) |
| Men | | | | | | | | | |
| 50–59 | 297 | 35 | 11.8 | 81 | 8 | 9.9 | 216 | 27 | 12.5 |
| 60–69 | 321 | 47 | 14.6 | 120 | 14 | 11.7 | 201 | 33 | 16.4 |
| 70–79 | 158 | 46 | 29.1 | 48 | 11 | 22.9 | 110 | 35 | 31.8 |
| 80–95 | 99 | 41 | 41.4 | 28 | 7 | 25.0 | 71 | 34 | 47.9 |
| Total population | 875 | 169 | 19.3 | 277 | 40 | 14.4 | 598 | 129 | 21.6 |
| Age-standardised prevalence (%)* | | | 17.0 (14.5 to 19.5) | | | 13.3 (9.3 to 17.3) | | | 18.7 (15.6 to 21.8) |
| P value for trend† | | | P<0.001 | | | P=0.013 | | | P<0.001 |
| Women | | | | | | | | | |
| 50–59 | 460 | 82 | 17.8 | 115 | 18 | 15.7 | 345 | 64 | 18.6 |
| 60–69 | 351 | 53 | 15.1 | 128 | 20 | 15.6 | 223 | 33 | 14.8 |
| 70–79 | 194 | 59 | 30.4 | 50 | 14 | 28.0 | 144 | 45 | 31.3 |
| 80–95 | 119 | 39 | 32.8 | 19 | 5 | 26.3 | 100 | 34 | 34.0 |
| Total population | 1124 | 233 | 20.7 | 312 | 57 | 18.3 | 812 | 176 | 21.7 |
| Age-standardised prevalence (%)* | | | 20.3 (17.9 to 22.6) | | | 18.5 (14.2 to 22.9) | | | 20.8 (18.0 to 23.6) |
| P value for trend† | | | P<0.001 | | | P=0.079 | | | P<0.001 |
| Both genders | | | | | | | | | |
| 50–59 | 757 | 117 | 15.5 | 196 | 26 | 13.3 | 561 | 91 | 16.2 |
| 60–69 | 672 | 100 | 14.9 | 248 | 34 | 13.7 | 424 | 66 | 15.6 |
| 70–79 | 352 | 105 | 29.8 | 98 | 25 | 25.5 | 254 | 80 | 31.5 |
| 80–95 | 218 | 80 | 36.7 | 47 | 12 | 25.5 | 171 | 68 | 39.8 |
| Total population | 1999 | 402 | 20.1 | 589 | 97 | 16.5 | 1410 | 305 | 21.6 |
| Age-standardised prevalence (%)* | | | 18.7 (17.0 to 20.4) | | | 15.9 (13.0 to 18.9) | | | 19.7 (17.7 to 21.8) |
| P value for trend† | | | P<0.001 | | | P=0.004 | | | P<0.001 |

*Estimated prevalence (95% CI) for projection by age-standardised to 2010 Chinese population census.
†P value for test of trend for age.

## Prevalence of URE

Visual impairment in the elderly is of increasing importance with longer life expectancy and the resultant growing senior population.[15] Refractive error is the most common cause of vision impairment when people age.[16] Since Schwab and Tielsch drew attention to the importance of correctable vision impairment, URE has gained increasing attention in recent years as a major cause of avoidable blindness and visual impairment.[17 18]

Given that refractive error is more common in East Asia, a great number of individuals with URE would be expected among the older Chinese population.[7] However, a great disparity in the prevalence of URE in older Chinese has been presented in previous population-based studies (table 3).[9 10 14 19 20] Our prevalence rate of URE is markedly higher than the rates in the Liwan Eye Study (7.0%; URE was defined as an improvement to 20/40 or better with automated refraction),[9] the Shihpai Eye Study (9.6%;

URE was defined as improving to better than 20/40 on refraction)[19] and the Hong Kong Study (13.4%; URE was defined as improving with pinhole to better than 20/60).[20] Although potential sources of errors exist in the recruitment ages and sampling methods, the discrepancies may be mainly due to differences in the definition of URE. Accordingly, the prevalence of URE in our study is similar to that in Singaporean-Chinese elders (21.7% in the Tanjong Pagar Study),[14] when the definition of URE is same. A previous study investigated in a rural block of Shanghai also showed a similar prevalence rate of URE (24.8% in the Baoshan Study) using a similar definition of URE.[10] Our finding expands the data suggesting that URE is a significant problem among older Chinese.

## Factors associated with URE

In the risk factor analysis, older age has been shown to be significantly associated with increasing risk of URE.

**Table 2** Logistic regression model of the predictors of uncorrected refractive error

| Variable | N | n (%) | Age OR (95% CI)* | P | Multivariate OR (95% CI)† | P |
|---|---|---|---|---|---|---|
| Age (per 1 year) | 1999 | 402 (20.1) | 1.05 (1.04 to 1.06) | **<0.001** | 1.04 (1.03 to 1.05) | **<0.001** |
| Gender | | | | 0.295 | | 0.675 |
| Male | 875 | 169 (19.3) | 1.0 | | 1.0 | |
| Female | 1124 | 233 (20.7) | 1.13 (0.90 to 1.41) | | 1.05 (0.84 to 1.32) | |
| Education | | | | **0.008** | | **0.019** |
| Secondary school and lower | 902 | 217 (24.1) | 1.36 (1.09 to 1.70) | | 1.34 (1.07 to 1.69) | |
| High school and above | 1097 | 185 (16.9) | 1.0 | | 1.0 | |
| Income | | | | 0.050 | | 0.156 |
| <5000 | 1327 | 292 (21.5) | 1.28 (1.00 to 1.64) | | 1.20 (0.93 to 1.55) | |
| 5000 and above | 672 | 110 (16.4) | 1.0 | | 1.0 | |
| History of ocular diseases | | | | **0.007** | | **0.009** |
| No | 1500 | 315 (21.0) | 1.0 | | 1.0 | |
| Yes | 499 | 87 (17.4) | 0.71 (0.55 to 0.91) | | 0.71 (0.55 to 0.92) | |

*Adjusted for age.
†Adjusted for age, gender, education, income and history of ocular diseases.
Bold values indicate statistically significant differences with a P-value less than 0.05.

In this study, the prevalence rate of URE increased with age from 15.5% in the 50–59 range to 36.7% in participants older than 80 years. This age-related trend was in accordance with previous population-based studies.[13 14 19 21–23] In addition, people with a low level of education were associated with URE. This is probably because a lower educational level may result in a lower socioeconomic status and a lack of awareness of refractive errors.[13 14 19 21 23] On the other hand, people with a history of ocular diseases were negatively associated with URE, which may be due to more ophthalmic services accessed than those without a history of ocular diseases. However, we could not confirm the previous association of URE with women.[13]

**Table 3** Prevalence rates and risk factors of uncorrected refractive error among Chinese population from the Jiangning and other eye studies

| Study (year of study) | Country/region | N | Age (years) | URE in study population (%) | Definitions (visual acuity (VA)) | Multivariate risk factors |
|---|---|---|---|---|---|---|
| Tanjong Pagar Survey[14] (1997–1998) | Singapore | 1152 | 40–79 | 21.7 | BCVA - PVA≥2 lines in the better eye of <20/40 | Older age, fewer years of education, not wearing spectacles, cataracts |
| Hong Kong Study[20] (1998) | Hong Kong | 3441 | 60+ | 13.4 | Improving with pinhole to better than 20/60 | NA |
| Shihpai Eye Study[19] (1999–2000) | Taiwan | 1361 | 65+ | 9.6 | Improving to better than 20/40 on refraction | Older age, non-emmetropic eye, not wearing spectacles, lower level of education |
| Liwan Eye Study[9] (2003–2004) | Guangzhou | 1399 | 50+ | 7.0 | Improvement to 20/40 or better with automated refraction | NA |
| Baoshan Study[10] (2009) | Shanghai | 4545 | 60+ | 24.8 | Improvement of two or more lines in VA in the better eye after refraction | NA |
| Jiangning Eye Study (2012–2013) (current) | Shanghai | 1999 | 50+ | 20.1 | BCVA - PVA≥2 lines in the better eye of <20/40 | Older age, secondary school and lower of education, no history of ocular diseases |

BCVA, best-corrected visual acuity; NA, not applicable; PVA, presenting visual acuity.

## Novel insight into the public health strategy

Both URE and cataract have been included in the priority areas of the global initiative Vision 2020: The Right to Sight to eliminate preventable blindness.[6] With more than 20% of the world's population residing in China alone, great efforts have been made in China to meet the goal of Vision 2020. According to the National Plan for the Prevention and Treatment of Blindness, the coverage of cataract surgery has increased significantly through projects such as Free Cataract Surgeries for A Million Poor Patients in China.[24] The cataract surgery rate in the Shanghai area increased from 1741 in 2006 to 4822 in 2016.[25 26] However, the findings of our study demonstrate the important contribution of URE to avoidable visual impairment in elderly Chinese aside from cataract and should raise awareness of the URE burden in China. An appropriate and cost-effective intervention for URE in older population should be considered during national health policy-making.

## Strengths and limitations of the study

The strengths of the present study include a high response rate in a large population-based sample, and standardised protocols based on the typical definition of URE. However, this study has several limitations. First, refractive error that impairs near vision, that is, presbyopia, was not assessed. Second, the gross domestic product per capita in Shanghai has approached the level of developed countries. All the subjects in the present study have health insurance, and ophthalmic consultations are easily accessed in this metropolitan area. We did not ask the reasons why they remained uncorrected for those subjects with URE in the present study, considering that economic factors and ophthalmic services should no longer be barriers. Further studies are needed to assess these issues. Furthermore, the present study was conducted in an urban population, which may limit the generalisability of the study findings. These results might not be extended to rural Chinese populations due to significant differences in social and economic factors between the urban and rural areas in mainland China.

## Conclusions

In summary, this study suggests that URE is highly prevalent among the elderly Chinese population in urban China. These data provide a novel insight into the public health strategy for the Vision 2020 initiative to prevent avoidable blindness and visual impairment in the world's most populous nation. Further investigation is needed to identify the effects of URE on the quality of life of the elderly and the underlying reasons why they remain uncorrected.

**Acknowledgements** The authors thank Jian Zhang (Zhongshan Ophthalmic Center, Sun Yat-sen University, Guangzhou, China) for providing suggestions on study sampling and statistical analysis.

**Contributors** HY and PZ conceived and designed the study. HY, YQ, XLiu, XC, WY and XLi performed the study. HY, YQ and QZ analysed the data. HY, YQ and PZ wrote the paper. All the authors read and approved the submitted version of the manuscript.

**Funding** This research received no specific grant from any funding agency in the public, commercial or not-for-profit sectors.

**Competing interests** None declared.

**Patient consent** Obtained.

**Ethics approval** Medical Ethics Committee of the Xinhua Hospital Affiliated to Shanghai Jiao Tong University School of Medicine (XHEC-C-2012-014).

**Provenance and peer review** Not commissioned; externally peer reviewed.

**Data sharing statement** No additional data are available.

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
