## [Reviewer comments · BMJ Open]

ARTICLE DETAILS

TITLE (PROVISIONAL)	Prevalence and Risk Factors of Uncorrected Refractive Error among an Elderly Chinese Population in Urban China: a Cross-sectional Study
AUTHORS	Ye, Hehua; Qian, Yiyong; Zhang, Qi; Liu, Xiaohong; Cai, Xuan; Yu, Wenjing; Li, Xiang; Zhao, Peiquan

VERSION 1 – REVIEW

REVIEWER	Marcelo Ayala Sahlgrenska Academy, Gothenburg University & Karolinska Institute. Sweden.
REVIEW RETURNED	10-Jan-2018

GENERAL COMMENTS	Dear authors, Thanks for the opportunity to revise your manuscript. I found it interesting. The study design seems to be good. Some small English grammar errors should be corrected. The information is not so new, patients without education (or economical resources) will not buy spectacles. A positive correlation with age is easy to understand, older patients are not working any longer and have no money or not necessity of spectacles. The authors have not described if the included subjects were operated, refractive or cataract surgery?
---

REVIEWER	Joshua Foreman Centre for Eye Research Australia, Australia
REVIEW RETURNED	11-Jan-2018

GENERAL COMMENTS	This is a concise and mostly well-written paper. My comments are very minor. Could the authors please fix up the following minor issues: Page 2. Line 21: Remove the word "comprehensive". the testing protocol that has been described is not comprehensive. Page 3. Line 18. Remove the word "of" Line 35. Please provide a reference Page 4. Line 11. Please add the word "people" after 78 million. Line 18. This statement that there have been few studies is not
---

	strictly true. Only 2 references are provided here, but the authors have provided 5 Chinese references in Table 3. Please remove this sentence. The authors should make the point here that although some studies have been conducted in various populations in China, the definitions of URE and prevalence estimates reported have varied between studies. Also, considering that there may be different risk factors and eye health services in China's heterogeneous population, prevalence rates of URE are likely to vary across the population. Quantifying the prevalence and risk in this large population is therefore warranted if Chinese health policy makers wish to optimize targeted strategies. Line 26. Change "determine" to "determining" Line 50. Change "approval" to "approved" Page 5. Line 30: Change "examination" to "examinations" Page 6. Line 33. Add "who" after "subjects" Line 40. Add "the" before "population" Line 45. Reword to "There was a high correlation between the refractive status of the right and left eyes" - Please also mention in the Statistical analysis section what statistical test was conducted to measure this correlation. Page 7. Line 16-18. The P value of 0.434 is far too high to make the claim that "The prevalence of URE in women (20.7%) was slightly higher than in men. They were very similar. Rather say that no significant difference was found. Line 23. Change "we" to "were" Page 8. Line 28. Add "the" before "elderly" Page 10. Line 38. An inappropriate symbol appears between "cost" and "effectiveness". Please remove. Page 11. Line 26. Change "quest" to "initiative"
--	---

VERSION 1 – AUTHOR RESPONSE

25-Jan-2018

Dear Dr. Hemali Bedi,

We would like to thank you and the reviewers for the constructive comments regarding our manuscript entitled "Prevalence and Risk Factors of Uncorrected Refractive Error among an Elderly Chinese Population in Urban China: The Jiangning Eye Study" (bmjopen-2017-021325). We have revised the manuscript according to the editors' and the reviewers' comments. Our point-to-point responses are shown below.

We hope that our revisions have appropriately addressed the editors' and the reviewers' concerns and that the manuscript is suitable for publication in BMJ Open.

Sincerely yours,

Peiquan Zhao, MD, PhD
Department of Ophthalmology
Xinhua Hospital, Shanghai Jiao Tong University School of Medicine
1665 Kongjiang Rd, Shanghai, 200092 P.R. China
Phone: 86-21-25078505
Fax: 86-21- 65011178
E-mail: zhaopeiquan@126.com; zhaopeiquan@xinhumed.com.cn

Editorial Requirements:

- Please revise your title to include the study design. This is the preferred format for the journal.

Response: Thank you very much. We have revised the title to "Prevalence and Risk Factors of Uncorrected Refractive Error among an Elderly Chinese Population in Urban China: a Cross-sectional Study".

Response to Reviewer 1:

Reviewer: 1

Reviewer Name: Marcelo Ayala

Institution and Country: Sahlgrenska Academy, Gothenburg University & Karolinska Institute. Sweden.

Please state any competing interests: None declared.

Please leave your comments for the authors below

Dear authors,

Thanks for the opportunity to revise your manuscript. I found it interesting. The study design seems to be good. Some small English grammar errors should be corrected. The information is not so new, patients without education (or economical resources) will not buy spectacles. A positive correlation with age is easy to understand, older patients are not working any longer and have no money or not necessity of spectacles.

The authors have not described if the included subjects were operated, refractive or cataract surgery?

Response:

Dear Dr. Marcelo Ayala,

We would like to thank you for the constructive comments regarding our manuscript. We have corrected the grammatical and spelling errors carefully, and had the manuscript proofread by a professional editing service (Scribendi Inc). The number of subjects with prior cataract surgery has been added in the results section of the revised manuscript. None of subject has a prior history of refractive surgery.

"Among the 101 participants with prior cataract surgery (at least one eye), 17.8% were uncorrected."
(Page 7)

Response to Reviewer 2:

Dear Dr. Joshua Foreman,

We would like to thank you for the invaluable comments regarding our manuscript. We have revised the manuscript accordingly. Our point-to-point responses are shown below.

Reviewer: 2

Reviewer Name: Joshua Foreman

Institution and Country: Centre for Eye Research Australia, Australia

Please state any competing interests: None declared

Please leave your comments for the authors below

This is a concise and mostly well-written paper. My comments are very minor. Could the authors please fix up the following minor issues:

Page 2.

Line 21: Remove the word "comprehensive". the testing protocol that has been described is not comprehensive. (Removed)

Page 3.

Line 18. Remove the word "of" (Removed)

Line 35. Please provide a reference (Added)

Page 4.

Line 11. Please add the word "people" after 78 million. (Added)

Line 18. This statement that there have been few studies is not strictly true. Only 2 references are provided here, but the authors have provided 5 Chinese references in Table 3. Please remove this sentence. The authors should make the point here that although some studies have been conducted in various populations in China, the definitions of URE and prevalence estimates reported have varied between studies. Also, considering that there may be different risk factors and eye health services in China's heterogeneous population, prevalence rates of URE are likely to vary across the population. Quantifying the prevalence and risk in this large population is therefore warranted if Chinese health policy makers wish to optimize targeted strategies.

Response: Thank you for your valuable comments and suggestion. In the manuscript, we use "China" to refer to "mainland China". The other 3 studies listed in Table 3 were performed in Chinese populations outside mainland China. We want to emphasize that few studies have reported the epidemiology of URE among the elderly in mainland China. In the revised manuscript, we have replaced "China" with "mainland China" in this paragraph, and have changed accordingly in Table 3. We hope that our revisions are acceptable.

Line 26. Change "determine" to "determining" (Changed)

Line 50. Change "approval" to "approved" (Changed)

Page 5.

Line 30: Change "examination" to "examinations" (Changed)

Page 6.

Line 33. Add "who" after "subjects" (Added)

Line 40. Add "the" before "population" (Added)

Line 45. Reword to "There was a high correlation between the refractive status of the right and left eyes" - Please also mention in the Statistical analysis section what statistical test was conducted to measure this correlation. (Added: The correlation between the refractive status of the right and left eyes was calculated using the Spearman correlation coefficient.)

Page 7. Line 16-18. The P value of 0.434 is far too high to make the claim that "The prevalence of URE in women (20.7%) was slightly higher than in men. They were very similar. Rather say that no significant difference was found. (Changed: No significant difference was found between men (19.3%) and women (20.7%) in the prevalence rate of URE ($\chi^2 = 0.613$, $P = 0.434$.)

Line 23. Change "we" to "were" (Changed)

Page 8.

Line 28. Add "the" before "elderly" (Added)

Page 10.

Line 38. An inappropriate symbol appears between "cost" and "effectiveness". Please remove.
(Removed)

Page 11.

Line 26. Change "quest" to "initiative" (Changed)

VERSION 2 – REVIEW

REVIEWER	Joshua Foreman Centre for Eye Research Australia, Australia
REVIEW RETURNED	30-Jan-2018

GENERAL COMMENTS	Suggested changes have been made.
-----------------------------------

REVIEWER	Marcelo Ayala Sahlgrenska Academy & Karolinska Institute, Sweden.
REVIEW RETURNED	31-Jan-2018

GENERAL COMMENTS	The authors have made necessary changes to the manuscript.
--